# Circulating Free DNA and Its Emerging Role in Autoimmune Diseases

**DOI:** 10.3390/jpm11020151

**Published:** 2021-02-20

**Authors:** Patricia Mondelo-Macía, Patricia Castro-Santos, Adrián Castillo-García, Laura Muinelo-Romay, Roberto Diaz-Peña

**Affiliations:** 1Liquid Biopsy Analysis Unit, Oncomet, Health Research Institute of Santiago (IDIS), 15706 Santiago de Compostela, Spain; patricia.mondelo.macia@sergas.es (P.M.-M.); laura.muinelo.romay@sergas.es (L.M.-R.); 2Inmunología, Centro de Investigaciones Biomédicas (CINBIO), Universidad de Vigo, 36310 Vigo, Spain; patricassan@gmail.com; 3Fissac—Physiology, Health, and Physical Activity, 28015 Madrid, Spain; adrian@fissac.com; 4Centro de Investigación Biomédica en Red de Cáncer (CIBERONC), 28029 Madrid, Spain; 5Faculty of Health Sciences, Universidad Autónoma de Chile, Talca 3460000, Chile

**Keywords:** liquid biopsy, cfDNA, autoimmune diseases, rheumatoid arthritis, systemic lupus erythaematosus, inflammatory bowel disease

## Abstract

Liquid biopsies can be used to analyse tissue-derived information, including cell-free DNA (cfDNA), circulating rare cells, and circulating extracellular vesicles in the blood or other bodily fluids, representing a new way to guide therapeutic decisions in cancer. Among the new challenges of liquid biopsy, we found clinical application in nontumour pathologies, including autoimmune diseases. Since the discovery of the presence of high levels of cfDNA in patients with systemic lupus erythaematosus (SLE) in the 1960s, cfDNA research in autoimmune diseases has mainly focused on the overall quantification of cfDNA and its association with disease activity. However, with technological advancements and the increasing understanding of the role of DNA sensing receptors in inflammation and autoimmunity, interest in cfDNA and autoimmune diseases has not expanded until recently. In this review, we provide an overview of the basic biology of cfDNA in the context of autoimmune diseases as a biomarker of disease activity, progression, and prediction of the treatment response. We discuss and integrate available information about these important aspects.

## 1. Introduction

Autoimmune diseases are multifactorial disorders characterized by the appearance of autoreactive immune cells and specific autoantibodies. There are >100 human diseases that are considered to be autoimmune or chronic inflammatory, and these diseases affect 5–10% of the world’s population [1]. The progressive evolution of knowledge of autoimmunity has depended on the use of antibodies as guides for pathogenesis, diagnosis, and prognosis [2]. Moreover, current research related to autoimmune diseases is mainly focused on different aspects of each condition: to identify novel biomarkers, to elucidate the mechanisms related to the etiopathogenesis of the disease, and to recognize or discover new therapeutic targets and agents. In this context, nucleic acid analyses are one of the most promising fields of research.

Based on the analysis of nuclear genomic material, our understanding of genetic susceptibility to autoimmune disease has substantially increased through genome-wide association studies (GWAS). These studies have driven the discovery of more than 300 susceptibility loci for autoimmune diseases [3], offering the possibility to identify individuals with a high risk for certain diseases [4]. In addition, the identification and analysis of circulating free DNA (cfDNA) has been described as a potential biomarker for autoimmune diseases. Although little is known about the precise function of cfDNA in autoimmunity, the advent of more sensitive methods and the increasing understanding of the role of DNA sensing receptors in inflammation and autoimmunity have clearly expanded interest in cfDNA analyses. CfDNA can also associate with different epigenetic modifications, being potential biomarkers in autoimmune diseases [5]. Therefore, in the present review, we provide an overview of the basic biology of cfDNA and the evolution of cfDNA in autoimmune diseases as a biomarker of disease activity, progression, and prediction of the treatment response. We discuss in detail and integrate available information about these important aspects, which are little explored.

## 2. Circulating Free DNA Characteristics and Clinical Interest

Serum, plasma, and other body fluids, such as urine, cerebrospinal fluids (CSFs), saliva, or bronchial effusions, are known to contain cfDNA, which represents a valuable biomarker in different clinical contexts, such as in prenatal diagnosis of genetic diseases, cancer detection and phenotyping to select personalized treatments, and cardiovascular diseases, among others [6]. It has been reported that cfDNA yields are higher in cancer patients than in healthy patients [7], but increased levels have also been reported in patients with benign lesions, inflammatory diseases, and tissue trauma [8,9].

### 2.1. Origin and Characteristics

CfDNA consists of a heterogeneous and complex DNA fraction present in free body fluids associated with extracellular vesicles (EVs) or as part of macromolecular complexes such as nucleosomes [10]. The size of cfDNA is highly variable (20–200 bp) depending on the mechanisms involved in its fragmentation, with a normal peak of 166 bp fragments, which corresponds to the length of the DNA bound to a nucleosome [6]. Due to this high fragmentation, cfDNA origin is mainly associated with cell death mechanisms; however, to date, the origin of cfDNA remains unclear, and different mechanisms have been suggested in several studies [11,12,13]. Here, we summarize the principal origins and sources of cfDNA described until now (Figure 1):

**Secretion**. Most cfDNA release into the circulation is associated with active secretion in EVs, such as exosomes, microparticles, or apoptotic bodies. This cfDNA is protected by nucleases and can be released into circulation through the breakdown of EVs. Some studies have reported that over 90% of cfDNA is associated with this type of release [14].

**Apoptosis**. Apoptosis, also known as programmed cell death, is an essential process to maintain cellular homeostasis. This process allows the removal of damaged cells by caspase activation. When the caspase pathway is activated, the cell starts to suffer morphological and biochemical changes that will result in cell and nuclear retraction, lipid redistribution, and DNA fragmentation. The cfDNA released by apoptosis is highly fragmented, double stranded, of low molecular weight, and approximately 150–200 bp in size [9].

**Necrosis**. Necrosis is an accidental cell death in response to physical or chemical injury characterized by cell swelling followed by loss of membrane integrity, with the consequent release of intracellular content. This process is more rapid than apoptosis, and there is no specific digestion of chromatin; therefore, the cfDNA obtained is larger, approximately 1000 bp [12].

**NETosis**. NETosis is a form of programmed cell death that neutrophils can undergo in response to microbes and sterile inflammation. The cfDNA fragments obtained via NETosis are similar to those obtained via necrosis [12].

**Pyroptosis**. Pyroptosis is an inflammatory process that induces the activation of inflammatory cytokines, interleukins and rapid cell death in response to diverse infections [15]. It has been reported to be closely associated with some diseases, such as atherosclerosis and diabetic nephropathy, and with cancer [16].

Recently, Aucamps et al., after an extensive bibliography review, concluded that cfDNA can arise from a single source but also from combinations of various sources and causes [12]. Moreover, several physiological processes, such as obesity [17], age [18], stress [19], and exercise [20], can induce the release of cfDNA into the circulation.

DNA can also be released from mitochondria into the cytoplasm and extracellular environments. Circulating mitochondrial DNA (mtDNA) was first reported by Zhong et al. in type 2 diabetes mellitus patients and healthy individuals [21]. Since then, many studies have reported the presence of mtDNA in the circulation in different diseases, such as cancer, autoimmune diseases, Parkinson’s disease, Alzheimer’s disease, and progressive multiple sclerosis, among others, including healthy individuals [12,22]. Due to the lack of histones and consequently of protection, the size of circulation-free mtDNA (cfmtDNA) is shorter (40–60 bp), being more fragmented than the total cfDNA. Importantly, cfmtDNA is not always correlated with total cfDNA levels, suggesting that cfmtDNA can be an independent and potential biomarker for some diseases [12].

### 2.2. Clearance of cfDNA

Some studies have investigated the variability of cfDNA across time and individuals [23], showing low individual fluctuations. Interestingly, age and sex in healthy individuals were associated with cfDNA concentration [24].

The cfDNA half-life varies from several minutes to 1–2 h [11,17,25]. It has been reported that their clearance is due to their degradation by some enzymes, such as deoxyribonuclease II and phosphodiesterase I. Additionally, cfDNA is eliminated from the circulation by organs such as the liver, spleen and kidney [9,11]. The liver was reported to play a major role in the clearance of cfDNA, while the kidney did not seem to be so involved in this process [17]. Therefore, the cfDNA level in blood is a result of a balance between cfDNA release and cfDNA clearance processes. In healthy individuals, cfDNA appears at low levels due to cells undergoing apoptosis and is rapidly removed. However, in malignancies, clearance of cfDNA is insufficient, and cfDNA levels increase. DNase I activity deficiency is one of the factors that can inhibit cfDNA degradation due to the association of cfDNA with proteins and enzyme recognition or the alteration of mechanisms regulating its activity [26].

In vivo models have reported that cfDNA originating from tumour cells (ctDNA) may elicit an inflammatory response in epithelial cells, suggesting that, under certain conditions, cfDNA can bypass protective mechanisms and be proinflammatory. There are two primary DNA-sensing pathways in cells that have been linked to cfDNA: the Toll-like receptor (TLR) 9 pathway and the stimulator of interferon genes (STING) pathway. These carrier proteins, often elevated in inflammatory conditions, can facilitate the uptake of DNA and protect the DNA from degradation, thus promoting the induction of proinflammatory responses [9].

### 2.3. Clinical Interest of cfDNA

CfDNA was first reported in healthy individuals by Mandel and Métais in 1948 [27]. However, it was not until 1966 when the discovery of high values of cfDNA in patients with systemic lupus erythaematosus (SLE) [28] showed the potential of cfDNA as a biomarker for autoimmune diseases. Ten years later, Leon et al. characterized cfDNA for the first time in the field of oncology and reported higher levels in cancer patients than in healthy individuals, suggesting its potential as a diagnostic marker and to characterize tumours in a noninvasive and dynamic way [7]. In 1994, cfDNA was recognized as an important tool to detect several mutations in the blood of patients with myeloid disorders [29] and pancreatic adenocarcinomas [30] and is currently a key element for precision oncology [31]. Levels of cfDNA originating from tumour cells (ctDNA) vary widely, representing more than 10% to 100% of the total cfDNA present in blood [32]. Diehl et al. reported that the ctDNA level correlates with tumour burden [33], and many other studies have demonstrated the value of ctDNA to monitor driver mutations and guide therapy [31,34].

Within recent years, the development of different kits based on the cfDNA analyses, and their approval as companion diagnostic tests for the management of oncologic patients, have opened a new avenue for the clinical application of ctDNA characterization. In 2016, the Food and Drug Administration (FDA) approved the first diagnostic test based on liquid biopsy: the Cobas^®^ EGFR Mutation Test v2 (Roche Molecular Systems, Inc., Pleasanton, CA, USA) based on the good results obtained in the clinical trial NCT01342965. The test allows the detection of epidermal growth factor receptor (*EGFR*) mutations using cfDNA from plasma samples in non-small cell lung cancer (NSCLC) patients for guiding the selection of patients who could benefit from anti-EGFR therapies [35]. In the same year, the Epi proColon^®^ test was the first FDA-approved blood-based colorectal cancer screening test, through the detection of methylated Septin9 DNA [36]. Therascreen PIK3CA RGQ PCR kit has also been approved by the FDA as a companion diagnostic for the detection of PIK3CA mutations in plasma samples from patients with advanced-stage breast cancer to determine their eligibility for blocking the PI3Kα mediated pathway [37]. Recently, two cfDNA-tests based on NGS technology have been also approved for a clinical use: the Guardant360 CDx (Guardant Health, Inc., Redwood City, CA, USA) in the context of NSCLC [38] and FoundationOne Liquid CDx test (Foundation Medicine, Inc., Cambridge, MA, USA) for patients with different solid malignant neoplasm [39]. Both panels allow the identification of patients who may benefit from treatments with targeted therapies.

Additionally, in the field of prenatal diagnosis, Lo et al. demonstrated the possibility of detecting circulating DNA of foetal origin (cffDNA) in the blood of pregnant women [40], showing a potential tool for the early identification of foetal genetic abnormalities such as aneuploidy using plasma samples [41].

Overall, several strategies have been developed to use cfDNA for the noninvasive screening of different diseases [12], including potential clinical applications for autoimmune diseases, inflammatory diseases, systemic disorders, trauma, sepsis, or myocardial infarction that are less known than the prenatal and oncology contexts [13].

## 3. Circulating Free DNA in Autoimmune Rheumatic Diseases

### 3.1. Systemic Lupus Erythaematosus

Systemic lupus erythaematosus (SLE) is an autoimmune disease with multiorgan damage, including damage to the skin, kidney, and joints and is characterized by the production of antibodies against nuclear antigens, named anti-nuclear antibodies (ANAs, Rome, Italy). Among these ANAs, the role of anti-double-stranded DNA (anti-dsDNA) antibodies has attracted great interest, as they are considered a specific marker for SLE [42]. Several potential mechanisms have been proposed to explain the pathogenic function of dsDNA [42], mainly from the perspective of kidney involvement [43]. The formation of immune complexes containing dsDNA and other nuclear antigens and their interaction with diverse pattern recognition receptors and internal sensing systems [44] could promote pathogenesis. Through the interaction with TLR9, anti-dsDNA complexes with DNA could determine the activation of dendritic cells, with consequent B and T-cell activation and proinflammatory cytokine release [45]. Besides, the appearance of antibodies against dsDNA and histones occurs together frequently, and some of them are even directed to histone-DNA complexes [46]. As result of the cell death mechanism and the associated inflammation, the intracellular or extracellular degradation of the dsDNA increases the histones levels, and their immunological activity can increase through the involvement of TLR9 [47]. In this context, the positioning of dsDNA/histones into the centre stage of SLE pathogenesis has attracted interest in the plasma and/or serum cfDNA of SLE patients.

High values of cfDNA in patients with SLE were first reported in the 1960s [28]. Since then, cfDNA research in SLE has mainly focused on its overall quantification and its association with disease activity. By the 1980s, serum DNA was found to be increased in patients with SLE and correlated with disease activity [48,49,50,51,52]. In parallel, conflicting data on the detection of cfDNA began to appear, concluding that the significance of serum and plasma cfDNA levels in SLE could be an artefact of the technology used for their characterization [53,54]. In 2007, Chen et al. optimized a fluorochrome PicoGreen assay for the ultrasensitive and reliable quantification of DNA in plasma/serum samples [55]. They concluded that for the most accurate quantification of cfDNA levels, it would be advisable to use plasma instead of serum samples. CfDNA levels detected in serum may be partially increased by nuclear DNA from leukocytes released during the clotting process due to the greater fragility or damage of the white blood cells from patients with SLE.

### 3.2. Rheumatoid Arthritis

Rheumatoid arthritis (RA) is a multifactorial, progressive, systemic, and inflammatory autoimmune disease that affects approximately 1% of the population worldwide [56]. Immediate and effective therapy is crucial to control inflammation and prevent deterioration, functional disability, and unfavourable progression in RA outcomes. Although several biomarkers routinely used in RA management, such as anti-citrullinated protein autoantibodies (ACPA) and rheumatoid factor (RF) tests, have greatly contributed to improving the early diagnosis of RA [57], there is a high demand for novel biomarkers to further improve not only the diagnosis but also the stratification of patients and even the prediction of response to a specific therapy. CfDNA appears to be a good candidate as a biomarker of early diagnosis of RA but also for disease monitoring and prediction of response to treatment [58].

Most of the studies have shown that cfDNA levels are higher in the plasma and serum of RA patients than in healthy controls, and there is also an association of cfDNA levels with disease activity and markers of inflammation [59,60,61,62]. Moreover, the detection and analysis of cfDNA and extracellular mtDNA in synovial fluid seems to show that cfDNA in RA patients is mainly located in joints [63,64,65] and is pathologically relevant. In these works, the cfDNA and cfmtDNA concentrations in synovial fluid were reported to be higher than the corresponding plasma levels.

### 3.3. Clinical Implication of cfDNA in Autoimmune Rheumatic Diseases

In recent years, high levels of cfDNA in SLE patients compared to healthy controls have been found [66,67]. In addition, SLE patients in an active phase of disease showed elevated cfDNA levels compared to patients with inactive disease [67]. This suggests that cfDNA levels might be a potential tool to assess and predict disease activity in patients with SLE, although further studies on a larger cohort of SLE patients are needed. In particular, potential clinical translation and the establishment of clinically significant reference ranges of cfDNA levels should be taken into account to define SLE patients with inactive and active disease. Changes in cfDNA levels might be one of the driving mechanisms behind flare-ups of SLE. Other fluids could also be used in order to analyse the cfDNA in SLE patients. The possibility of using urine to analyse the cfDNA in cancer patients has been reported [68]. In SLE, due to the production of urine in the kidneys, it could be an alternative fluid sample, especially in patients with lupus nephritis (LN). LN is the most common complication and cause of kidney injury in SLE patients. Nowadays, their diagnosis is based on a renal biopsy, although several urinary biomarkers, such as miRNAs, have been reported [69,70], showing the potential of the urine in the management of this disease.

SLE can cause inflammation and damage to various tissues in a chronic manner and cell death, causing the release of cfDNA. CfDNA itself may perpetuate ongoing inflammation, forming a loop where a treatment that reduces systemic inflammation probably also affects cfDNA levels. Different types of programmed apoptosis, such as pyroptosis and NETosis, could explain how nucleic acids engage intracellular receptors and stimulate inflammation [71]. However, it is necessary to clarify the association of inflammation with higher cfDNA levels in SLE patients, which has not been clear in the literature until now [72].

Regarding RA, Dong et al. showed that synovial fluid cfDNA is inflammatogenic and facilitates the potent induction of inflammatory mediators that are critical for RA pathogenesis [65] (Figure 2). Molecular analysis in this study revealed that synovial fluid cfDNA is enriched with specific hypomethylated CpG motif-rich sequences that were previously shown to have a strong proinflammatory capability in RA-related cells [73,74]. This provides a novel target for treatment and a potential biomarker for RA. There is no evidence that the role of cfDNA might be related to the presence of ACPA and RF, and the data are contradictory [59,61]. This allows us to think that cfDNA could be an independent biomarker of ACPA/RF, being able to provide added value in a combined manner.

Few studies analysing the association between the changes in cfDNA concentration in RA patients and treatment with biological disease-modifying anti-rheumatic drugs (bDMARDs) have been reported. Hashimoto et al. showed that cfDNA was able to predict the therapeutic effects of bDMARDs in RA patients [62]. This work described an increase in cfDNA 8 weeks after introducing bDMARDs associated with an improvement in disease activity. However, in a subsequent study, Lauková et al. showed that plasma cfDNA decreases during dDMARD therapy [75], for which no clear conclusions can be drawn. Moreover, there is an important challenge in the selection of antitumour necrosis factor (TNF) drugs, where individual patients show great variability in response [76]. However, anti-TNF-based treatment strategies do not appear to alter cfDNA concentrations [60]. Taking into account all these data, further research in larger cohorts of patients is needed to evaluate the potential of cfDNA dynamics in RA as an indicator of the response to therapy.

Concerning other autoimmune rheumatic diseases there are barely any data. It would be interesting to investigate the role of cfDNA in spondyloarthropathies (SpA), a diverse group of chronic inflammatory conditions linked by distinctive clinical, radiographic, and genetic features. The SpA include ankylosing spondylitis (AS), psoriatic arthritis (PsA), reactive arthritis (ReA), enteropathic or inflammatory bowel disease (IBD)-associated SpA, and undifferentiated SpA. Leon et al. analyzed the DNA levels in paired samples of serum and synovial fluid from patients with arthritis, including some samples from patients with psoriatic arthropathy and AS, but no clear results were found [63]. Recently, Birkelund et al. reported that the presence of cfDNA correlates with proteins predominantly found in neutrophil granulocytes in synovial fluid from SpA patients [77]. Further studies involving larger numbers of SpA patients are needed to investigate the role of cfDNA in these kinds of diseases.

## 4. Circulating Free DNA in Inflammatory Bowel Disease

Due to the potential role of cfDNA in inflammatory processes and therefore in autoimmune diseases, particularly in SLE and RA, its function in the pathogenesis of inflammatory bowel disease (IBD) has also been suggested [78]. IBD represents a multifactorial chronic inflammatory process affecting the gastrointestinal tract [79] and includes Crohn’s disease (CD) and ulcerative colitis (UC). Although they share many similarities, CD and UC represent different diseases, displaying diverse therapeutic responses. However, the hallmark of both diseases is inflammation.

Increased excretion of human DNA has been observed in patients with active UC [80,81]. Vincent et al. reported that intestinal inflammation can occur prior to *Clostridium difficile* infection (CDI) development [82], so the quantification of human DNA in faeces could serve as a simple and noninvasive approach to assess bowel inflammation and identify patients at risk of CDI. This would suggest that the faecal DNA concentration might act as an index of mucosal inflammation and damage. Different authors have also detected significantly higher levels of plasma cfDNA in mice with dextran sulfate sodium (DSS)-induced colitis compared to the control group [83,84], suggesting that colon cells could represent a major source of colonic plasma cfDNA, being able to initiate events that contribute to inflammation and pathogenesis. The innate immune system is a crucial factor in understanding the pathogenesis of IBD [85,86], which leads to the activation of the DNA-sensing pathways TLR9 and STING. Some studies have already explored the role of these pathways in IBD. For example, TLR9 is an important element of protection against intestinal damage and for intestinal repair [87], and its stimulation could play a relevant role in regulating intestinal homeostasis, being a potential therapeutic target to enhance intestinal wound repair in IBD. However, this effect could be mediated by bacterial cfDNA derived from intestinal microbiota [88], which could also play diverse roles in the pathogenesis of IBD [89]. Recently, Boyapati et al. showed that significantly increased levels of cfmtDNA are found in active human IBD and in mouse colitis [90], suggesting cfmtDNA-TLR9 signalling as an important and targetable pathway in IBD. In fact, it has been described that cfmtDNA can activate various inflammatory responses via TLR9 receptors, including NLRP3 inflammasomes and neutrophils, together with other pro-inflammatory signaling pathways regulated by NFκB and TNFα [90,91]. Further research is needed to clarify the role of each element, even their potential interactions.

## 5. Circulating Free DNA in Other Autoimmune Disorders

Psoriasis is a multifactorial chronic inflammatory disease characterized by the activation of keratocytes due to dysregulation of the cellular immune system. Similar to the other diseases described above, the presence of cfDNA, both in serum and plasma, can reflect the cellular damage in this disease coming from apoptotic or necrotic cells [92,93]. Recently, Sakamoto et al. investigated whether the TNF-α gene was present in cfDNA and whether its levels could be used as a biomarker in psoriasis [94]. TNF-α is a key proinflammatory cytokine that activates keratocytes, exacerbating an inflammatory process and constituting an inseparable element of psoriasis [95]. The authors suggested that the levels of TNF-α copies in cfDNA might be a biomarker for severity in psoriasis patients, but further larger studies are needed to check its clinical potential. The same needs apply to other autoimmune diseases, featuring very preliminary data. For example, in celiac disease (CD), evidence that plasma cfDNA of patients has different immunoregulatory properties than plasma of healthy controls has been reported [96]; however, there are no differences in the cfDNA concentration between both groups. On the other hand, type 1 diabetes (T1D) is an autoimmune disease characterized by the destruction of insulin-producing beta cells of the pancreatic islets of Langerhans, producing chronic inflammation and a progressively severe insulin deficit [97]. Dying beta cells during the pathology of T1D release their DNA content into the circulation. The measurement of beta cell cfDNA, including insulin, using next-generation sequencing (NGS) or digital droplet PCR (ddPCR) is a promising approach to determine the methylation status in plasma or serum [98]. Another study suggests a correlation among the methylation patterns of cfDNA in recently diagnosed T1D patients, the rate of beta-cell death, and the risk of developing the disease [99]. In T1D patients with difficulties in controlling their blood glucose levels through insulin management, the replacement of beta-cells (through whole pancreas transplantation or islet transplantation) represents a therapeutic option [100]. The analysis of methylation in cfDNA from T1D patients after the replacement of beta-cells correlates with clinical outcomes, thus predicting early engraftment [101]. Together with other clinical indicators, this type of analysis could contribute to ensuring long-term graft survival by allowing timely interventions and better planning of subsequent islet infusions. Sjögren’s syndrome (SS) patients also show high cfDNA levels in serum, with a significant correlation between cfDNA concentrations and disease activity [102]. Recently, Vakrakou et al. reported a systemic NLRP3 inflammasome activation in severe SS, associated with widespread extranuclear accumulations of inflammagenic DNA and impaired DNA degradation [103]. These data suggest a potential role of cfDNA as non-invasive marker of SS activity.

On the other hand, CSF represents an accessible source of central nervous system (CNS)-derived products, which can reflect molecular changes in the CNS. In this sense, the potential role of cfmtDNA levels in multiple sclerosis (MS) has been studied. MS is a chronic inflammatory disorder of the CNS, where mitochondrial dysfunction is recognized as an important feature of this pathology [104]. An increase in cfmtDNA copies in both relapsing-remitting MS (RRMS) and progressive MS (PMS) patients has been reported [105,106,107]. However, recently, Lowes et al. reported a decrease in cfmtDNA in postmortem ventricular CSF (vCSF) in 36 PMS patients, concluding that this decrease may be an indicator of neurodegeneration in PMS [22]. Ligget et al. reported an increase in the cfDNA concentration in RRMS patients compared to healthy controls in plasma, showing unique disease-specific gene promoter methylation patterns [108]. Similarly, the role of serum cfDNA in MS patients has been studied, and different patterns of methylation between RRMS patients and healthy controls have been reported, thus suggesting that the methylation status of cfDNA may reflect pathophysiological phenomena in the brain [109,110].

## 6. Conclusions and Future Perspectives

CfDNA represents a relevant noninvasive tool for personalized medicine. There is evidence to consider cfDNA as a potential biomarker in autoimmune diseases, at least in SLE and RA. However, there are different aspects that need to be addressed to validate its use in the context of autoimmune diseases. Technical problems related to the technique’s homogenization must be solved in the close future, such as the type of sample used for the analysis (plasma and/or serum, even synovial fluid, or CFS), sample collection (tubes) and processing, or cfDNA extraction and quantification. For example, CSF might be an optimal source to analyse cfmtDNA in MS patients [105,106]. The development of some autoimmune diseases is accompanied by changes in cfDNA levels that can be pathologically relevant. It seems clear, but further research is needed to define the specific role of cfDNA in each of the diseases.

Overall, DNA methylation also seems to be a promising alternative for early diagnosis and monitoring of the autoimmune diseases [111]. The analysis of changes in cfDNA methylation levels suggest potential to stratify patients depending on different stages in the diseases, but could also serve to monitor the response to specific treatments [5]. By using tissue/disease-specific epigenetic marks on the cfDNA, it might be possible to offer personalized medicine to manage the autoimmune diseases in the future.

During the past decade, cfDNA has received much attention in the field of oncology [112]. The capacity and sensitivity of cfDNA analyses has increased due to the development of new approaches and the improvement of technologies such as ddPCR and NGS. Translation to complex traits, autoimmune diseases in this case, will take more time, as our understanding of somatic alterations and evolution in normal tissues is limited. Serra et al. reported a case study of a very-early-onset IBD patient with a mosaic de novo pathogenic allele in CYBB via whole-exome sequencing [113], showing that this mutation was present in ~70% of phagocytes and sufficient to result in defective bacterial handling but not life-threatening infections. Recently, Olafsson et al. provided the most accurate characterization of the somatic mutation landscape of the IBD-affected colon to date through whole-genome sequencing of individual colonic crypts [114]. They suggested that somatic evolution in the colonic mucosa may initiate, maintain, or perpetuate IBD. This represents an important step forward since different selection mechanisms in the colitis-affected colon and somatic mutations could potentially play a causal role in IBD pathogenesis. The extension of this type of study to other tissues related to autoimmune diseases could be very interesting, expanding an area of research that is still in its only beginning. It is clear that the study of somatic mutations in nonneoplastic diseases lags many years behind the study of mutations in cancer. If cfDNA can be as useful in complex diseases as cancers, it is an issue that can generate interesting research hypotheses in the future. As the basis, it would be necessary to know whether a pathogenic clone exists and what mutations it carries, even if those mutations have an effect on treatments.

The value of cell-free DNA as a therapeutic target in autoimmune disease has gained interest. For example, a low DNase I activity was described in some disorders such as IBD. The treatment with DNase was shown to be active in several immune-mediated experimental models through the breaking of NETs and the consequent pro-inflammatory activity [115,116]. Another promising line of work is focused on the development of nucleic acid-binding polymers (NABPs), which can scavenge proinflammatory cfDNA to modulate inflammation at the injured site [117,118]. Of note, the therapeutic potential of cfDNA in this context of research should be further explored in the near future.

Overall, cfDNA has been widely studied in cancer. We have seen that cfDNA could also be a useful tool in the autoimmune disease field. However, further studies are needed to elucidate standardized methods to analyse the cfDNA in these patients. CfDNA might be a biomarker of disease activity, progression, and prediction of the therapy response in SLE, RA, and IBD. Moreover, the role of cfDNA in other autoimmune diseases, such as CD, psoriasis, T1D, and MS, is starting to be studied.

## Figures and Tables

**Figure 1 jpm-11-00151-f001:**
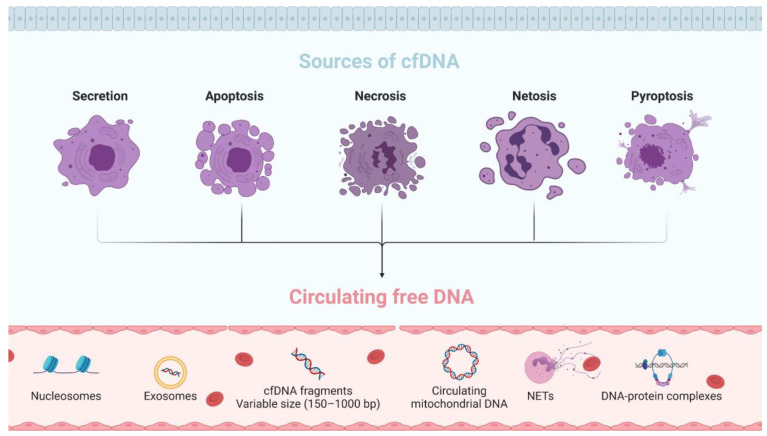
Mechanisms involved in the release of cell-free DNA.

**Figure 2 jpm-11-00151-f002:**
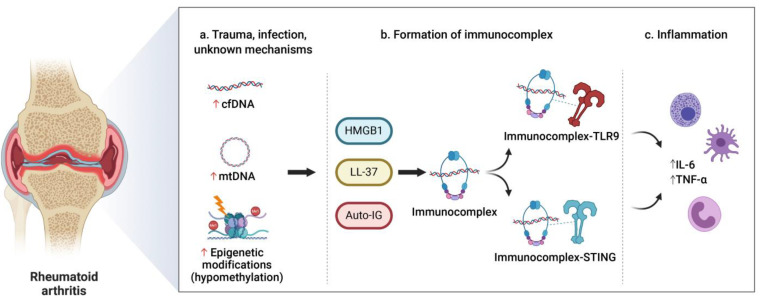
Potential mechanisms involving synovial fluid circulating-free DNA and the development of rheumatoid arthritis (RA). Release of cfDNA could increase in conditions including trauma, infections, or other mechanisms that we do not know (**a**). In addition, synovial fluid cfDNA seems to be enriched with specific hypomethylated CpG motif-rich sequences, with capacity to induce severe inflammatory responses independently. CfDNA and mtDNA can bind with LL-37, HMGB1, auto-Ig, and other proteins to form immunocomplexes that can be recognized by pattern recognition receptors (TLR-9 or STING) (**b**). This would facilitate the induction of cytokine overexpression, TNF-α and IL-6, that are critical for RA pathogenesis (**c**). Abbreviations: cfDNA, circulating-free DNA; HMGB1, high mobility group box chromosomal protein 1; IG, immunoglobulins; IL-6, interleukin 6; LL-37, human cathelicidin LL-37; mtDNA, mitochondrial DNA; STING, stimulator of interferon genes; TLR-9, toll-like receptor 9; TNF-α, tumor necrosis factor alpha.

## Data Availability

The data presented in this study are available on request from the corresponding author.

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
