# Peer review of "Circulating Free DNA and Its Emerging Role in Autoimmune Diseases"

_jpm, 2021, doi:10.3390/jpm11020151_

Round 1

Reviewer 1 Report

Current manuscript of Mondelo-Macia et al. is an average quality review of a quite relatively novel topic.

Nowhere, I think it should be necessary some MAJOR REVISIONS before considering acceptance by Editors.

Among POINTs of WEAKNESSES we can list:

  • An important re-check of English language (i.e. sentences as at lines 33-34 might be re-written) such as some minor spelling errors (i.e. at line 285) 
  • A further extension of current DISCUSSION by means of some possibile suggested readings like:
    • Lancet. 2018 Sep 1;392(10149):777-786. doi: 10.1016/S0140-6736(18)31268-6. Epub 2018 Aug 9
    • Autoimmun Rev. 2016 Jul;15(7):756-69. doi: 10.1016/j.autrev.2016.03.014. Epub 2016 Mar 12.
      • FOR INTRODUCTION IMPROVEMENT
    • Front Genet. 2020 Aug 11;11:844. doi: 10.3389/fgene.2020.00844. eCollection 2020.
      • FOR ENLARGING DISCUSSION
  • Finally, a quality improvement of figure 1 and/or a consistent ameliorating of figure 2 caption.

Best regards.

Author Response

  1. An important re-check of English language (i.e. sentences as at lines 33-34 might be re-written) such as some minor spelling errors (i.e. at line 285)

R: Our apologies for this. Manuscript have been rechecked with the assistance of "American Journal Experts" (https://www.aje.com/). Attached Editing Certificate.

  1. A further extension of current Introduction and Discussion by means of some possible suggested readings

R: Thank you for the comment. According to the Reviewer’s suggestion, we have included and discussed the recommended references (page 2, lines 52-53; and page 9, lines 442-447).

  1. A quality improvement of figure 1 and/or a consistent ameliorating of figure 2 caption.

R: We thank the reviewer suggestion. We have increased the quality of figure 1, and renamed the figure 2 caption. We attached the figures 1 and 2 independently, in high quality.

Reviewer 2 Report

Dear Authors,

Although the review manuscript is quite comprehensive, i would suggest the addition of a section on in-clinical development of cfDNA based kits/diagnostics. Apart from this recommendation the review is of publication quality.

Good luck

Author Response

  1. I would suggest the addition of a section on in-clinical development of cfDNA based kits/diagnostics.

R: According to the Reviewer’s suggestion, we have included a section on in-clinical development of cfDNA (page 4, lines 180-198), adding 5 new references (35-39)

Reviewer 3 Report

In this review, authors provide an overview of the basic biology of cell-free (cf)DNA and the evolution of cfDNA in autoimmune diseases as a biomarker of disease activity, progression, and prediction of the therapeutic response by integrating the available information. Some comments are as the follows:  

  1. A major concern is that a proportion of review content is similar to the published review manuscript [Duvvuri B et al. Cell-freee DNA as a biomarker in autoimmune rheumatic diseases. Front Immunology 2019]. It would be better that authors could include other inflammatory diseases such as auto-inflammatory diseases.
  2. It would be helpful if the authors could summarize the clinical implication of cfDNA in autoimmune rheumatic diseases in a separate section. Furthermore, it would be better to illustrate it in figure.
  3. The role or pathogenic mechanisms of cfDNA could be presented in more detail. Some important findings are not shown in this review.
  4. The cfDNA in urine samples may be an important issue for patients with lupus nephritis. The authors may add into this review.   

5. It would be better that author could add the limitations and challenges.

Author Response

  1. A major concern is that a proportion of review content is similar to the published review manuscript [Duvvuri B et al. Cell-freee DNA as a biomarker in autoimmune rheumatic diseases. Front Immunology 2019]. It would be better that authors could include other inflammatory diseases such as auto-inflammatory diseases. It would be helpful if the authors could summarize the clinical implication of cfDNA in autoimmune rheumatic diseases in a separate section. Furthermore, it would be better to illustrate it in figure.

R: Our goal, from the beginning, was always to bring together the experience of the authors in liquid biopsy and autoimmunity. In fact the work of Duvvuri B et al. has been a key piece in our manuscript, particularly with SLE and RA. However, we have explored the possible and emerging role of cfDNA in other important autoimmune disease (IBD, psoriasis, celiac disease, type I diabetes, multiple sclerosis), not previously reviewed and integrated in the same work. Moreover, thanks to the reviewer comments, we have added and expanded the following parts of the manuscript.

- We have grouped the autoimmune rheumatic diseases (SLE and RA) into one section, adding some comments regarding to the spondyloarthropathies (page 7, lines 335-346).

- We have expanded the Introduction and Discussion, with bibliography on the possible role of the association cfDNA-epigenetic modifications in autoimmune diseases (page 2, lines 52-53; and page 9, lines 442-447).

- We have included data related to Sjögren's syndrome and cfDNA in the "Circulating free DNA in other autoimmune disorders" section (pages 8-9, lines 409-414).

- In relation to auto-inflammatory diseases, we have not found relevant data suggesting a potential role of cfDNA.

- In figure 2, we only illustrate the potential mechanisms by which cfDNA might be implicated in onset and development of RA. We discard include SLE, for example, due to the redundancy of the mechanisms, and above all for the uniqueness of the synovial fluid. However, we are open to make any kind of modification or addition.

  1. The role or pathogenic mechanisms of cfDNA could be presented in more detail. Some important findings are not shown in this review.

R: We appreciate the comment of the reviewer on the relevance of describing the pathogenic role of cfDNA in the different autoimmune diseases included in the review, therefore we comment the most validated knowledge about this point for the different autoimmune disorders including new information and that explained in Figure 2 (page 5, lines 213-226; page 6, lines 277-281; page 8, lines 367-378).

  1. The cfDNA in urine samples may be an important issue for patients with lupus nephritis. The authors may add into this review.

R: According to the Reviewer’s suggestion, we have commented the possibility to analyze cfDNA in urine samples in patients with lupus nephritis (page 6, lines 268-275), adding 3 new references (68-70)

  1. It would be better that author could add the limitations and challenges.

R: Thank you for the comment. We have rewritten the Section 6 “Conclusions and future perspectives” of the manuscript, in order to highlight the limitations and challenges and included a new comment regarding the therapeutic opportunities of cfDNA in autoimmune disorders. Please see revised manuscript.

Round 2

Reviewer 1 Report

I have much appreciated efforts of authors and modifications to the paper as suggested and so now I think it is suitable for publication by editors.

Reviewer 3 Report

Although authors cannot address all my comments such as illustration in figuring, I do not have any further comment.